# 3D Concept Grounding on Neural Fields

**Yining Hong**
University of California, Los Angeles

**Yilun Du**
Massachusetts Institute of Technology

**Chunru Lin**
Shanghai Jiao Tong University

**Joshua B. Tenenbaum**
MIT BCS, CBMM, CSAIL

**Chuang Gan**
UMass Amherst and MIT-IBM Watson AI Lab

## Abstract

In this paper, we address the challenging problem of 3D concept grounding (*i.e.* segmenting and learning visual concepts) by looking at RGBD images and reasoning about paired questions and answers. Existing visual reasoning approaches typically utilize supervised methods to extract 2D segmentation masks on which concepts are grounded. In contrast, humans are capable of grounding concepts on the underlying 3D representation of images. However, traditionally inferred 3D representations (*e.g.*, point clouds, voxelgrids and meshes) cannot capture continuous 3D features flexibly, thus making it challenging to ground concepts to 3D regions based on the language description of the object being referred to. To address both issues, we propose to leverage the continuous, differentiable nature of neural fields to segment and learn concepts. Specifically, each 3D coordinate in a scene is represented as a high dimensional descriptor. Concept grounding can then be performed by computing the similarity between the descriptor vector of a 3D coordinate and the vector embedding of a language concept, which enables segmentations and concept learning to be jointly learned on neural fields in a differentiable fashion. As a result, both 3D semantic and instance segmentations can emerge directly from question answering supervision using a set of defined neural operators on top of neural fields (*e.g.*, filtering and counting). Experimental results on our collected PARTNET-REASONING dataset show that our proposed framework outperforms unsupervised / language-mediated segmentation models on semantic and instance segmentation tasks, as well as outperforms existing models on the challenging 3D aware visual reasoning tasks. Furthermore, our framework can generalize well to unseen shape categories and real scans[*].

## 1 Introduction

Visual reasoning, the ability to utilize compositional operators to perform complex visual question answering tasks (*e.g.,* counting, comparing and logical reasoning), has become a challenging problem these years since they go beyond pattern recognition and bias exploitation [19, 22, 2]. Consider an image of a table pictured in Figure 1. We wish to construct a method that is able to accurately ground the concepts and reason about the image such as the number of legs the pictured table has. Existing works typically address this problem by utilizing a supervised segmentation model for legs, and then applying a count operator on extracted segmentation masks [28]. However, as illustrated in Figure 1, in many visual reasoning problems, the correct answer depends on a very small portion of

---

[*]Project page: http://3d-cg.csail.mit.edu

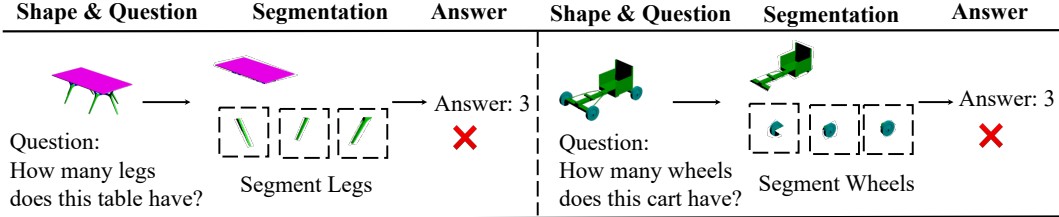

| Shape & Question | Segmentation | Answer | Shape & Question | Segmentation | Answer |

Figure 1: **Issues with 2D Concept Grounding and Visual Reasoning.** Existing methods typically answer questions by relying on 2D segmentation masks obtained from supervised models. However, heavy occlusion leads to incorrect segmentations and answers.

a given image, such as a highly occluded back leg, which many existing 2D segmentation systems may neglect. Humans are able to accurately ground the concepts from images and answer such questions by reasoning on the underlying 3D representation of the image [43]. In this underlying 3D representation, the individual legs of a table are roughly similar in size and shape, without underlying issues of occlusion of different legs, making reasoning significantly easier and more flexible. In addition, such an intermediate 3D representation further abstracts confounding factors towards reasoning, such as the underlying viewing direction from which we see the input shape. To resemble more human-like reasoning capability, in this paper, we propose a novel concept grounding framework by utilizing an intermediate 3D representation.

A natural question arises – what makes a good intermediate 3D representation for concept grounding? Such an intermediate representation should be discovered in a weakly-supervised manner and efficient to infer, as well as maintain correspondences between 3D coordinates and feature maps in a fully differentiable and flexible way, so that segmentation and concept learning can directly emerge from this inferred 3D representation with supervision of question answering. While traditional 3D representations (*e.g.*, point clouds, voxelgrids and meshes) are efficient to infer, they can not provide continuous 3D feature correspondences flexibly, thus making it challenging to ground concepts to 3D coordinates based on the language descriptions of objects being referred to. To address both issues, in this work, we propose to leverage the continuous, differentiable nature of neural fields as the intermediate 3D representation, which could be easily used for segmentation and concept learning through question answering.

To parameterize a neural field for reasoning, we utilize recently proposed Neural Descriptor Fields (NDFs) [41] which assign each 3D point in a scene a high dimensional descriptor. Such descriptors are learned in a weakly-supervised manner, and implicitly capture the hierarchical nature of a scene, by leveraging implicit feature hierarchy learned by a neural network. Portions of a scene relevant to a particular concept could be then differentiably extracted through a vector similarity computation between each descriptor and a corresponding vector embedding of the concept, enabling concept segmentations on NDFs to be differentiably learned. In contrast, existing reasoning approaches utilize supervised segmentation models to pre-defined concepts, which prevents reasoning on concepts unseen by the segmentation model [28], and further prevents models from adapting segmentations based on language descriptions.

On top of NDFs, we define a set of neural operators for visual reasoning. First, we construct a filter operator, and predict the existence of a particular concept in an image. We also have a query operator which queries about an attribute of the image. In addition, we define a neural counting operator, which quantifies the number of instances of a particular concept.

To evaluate the performances of our 3D Concept Grounding (3D-CG) framework, we conduct experiments on our collected PARTNET-REASONING dataset, which contains approximately 3K images of 3D shapes and 20K quesion-answer pairs. We find that by integrating our neural operators with NDF, our framework is able to effectively and robustly answer a set of visual reasoning questions. Simultaneously, we observe the emergence of 3D segmentations of concepts, at the semantic and instance level, directly from the underlying supervision of downstream visual question answering.

Our contributions can be summed up as follows: 1) we propose 3D-CG, which utilizes the differentiable nature of neural descriptor fields (NDF) to ground concepts and perform segmentations; 2) we define a set of neural operators, including a neural counting operator on top of the NDF; 3) with

3D-CG, semantic and instance segmentations can emerge from question answering supervision; 4) our 3D-CG outperforms baseline models in both segmentation and reasoning tasks; 5) it can also generalize well to unseen shape categories and real scans.

## 2 Related works

**Visual Reasoning.** There have been different tasks focusing on learning visual concepts from natural language, such as visually-grounded question answering [11, 12] and text-image retrieval [45]. Visual question answering that evaluates machine's reasoning abilities stands out as a challenging task as it requires human-like understanding of the visual scene. Numerous visual reasoning models have been proposed in recent years. Specifically, MAC [16] combined multi-modal attention for compositional reasoning. LCGN [15] built contextualized representations for objects to support relational reasoning. These methods model the reasoning process implicitly with neural networks. Neural-symbolic methods [53, 28, 3] explicitly perform symbolic reasoning on the objects representations and language representations. Specifically, they use perception models to extract 2D masks for 3D shapes as a first step, and then execute operators and ground concepts on these pre-segmented masks, but are limited to a set of pre-defined concepts. In this paper, we present an approach towards grounding 3D concepts in a fully differentiable manner, with which 3D segmentations can emerge from question answering supervision.

**Neural Fields.** Our approach utilizes neural fields also known as neural implicit representations, to parameterize an underlying 3D geometry of a shape for reasoning. Implicit fields have been shown to accurately represent shape topology and 3D geometry [1, 20, 32]. Recent works improve traditional shape fields by using continuous neural networks [33, 5, 36, 38, 39, 51, 13]. These works are typically used for reconstruction [33, 36, 29] or geometry processing [51, 37, 8]. We use a neural descriptor field (NDF) similar to [29] to infer the 3D representations from 2D images, and leverage the features learnt from the NDF for visual reasoning and language grounding. Neural fields have also been used to represent dynamic scenes [34, 10], appearance [42, 30, 35, 52, 40], physics [23], robotics [18], acoustics [27] and more general multi-modal signals [9]. Conditional neural field such as [25, 48] allows us to encode instance-specific information into the field, which is also utilized by our framework to encode the RGBD data of 3D shapes. There are also some works that integrate semantics or language in neural fields [17, 46]. However, they mainly focus on using language for manipulation, editing or generation. In this work, we utilize descriptors defined on neural fields [41] to differentiably reason in 3D given natural language questions. We refer readers to recent surveys [49, 44] for more related works on neural fields.

**Language-driven Segmentation.** Recent works have been focused on leveraging language for segmentation. Specifically, BiLD [14] distills the knowledge from a pre-trained image classification model into a two-stage detector. MDETR [21] is an end-to-end modulated detector that detects objects in an image conditioned on a raw text query, like a caption or a question. LSeg [24] uses a text encoder to compute embeddings of descriptive input labels together with a transformer-based image encoder that computes dense per-pixel embeddings of the input image. GroupViT [50] learns to group image regions into progressively larger arbitrary-shaped segments with a text encode. These methods typically use an encoder to encode the text and do not have the ability to modify the segmentations based on the question answering loss. LORL [47] uses the part-centric concepts derived from language to facilitate the learning of part-centric representations. However, they can only improve the segmentation results but cannot generate segmentations from scratch with language supervision.

## 3 3D Concept Grouding

In Figure 2, we show an overall framework of our 3D Concept Grounding (3D-CG), which seamlessly bridges the gap between 3D perception and reasoning by using Neural Descriptor Field (NDF). Specifically, we first use NDF to assign a high dimensional descriptor vector for each 3D coordinate, and run a semantic parsing module to translate the question into neural operators to be executed on the neural field. The concepts, also the parameters of the operators, possess concept embeddings as well. Upon executing the operators and grounding concepts in the neural field, we perform dot product attention between the descriptor vector of each coordinate and the concept embedding vector to calculate the score (or mask probability) of a certain concept being grounded on a coordinate. We also propose a count operator which assigns point coordinates into different slots for counting the number of instances of a part category. Both the NDF and the neural operator have a fully differentiable

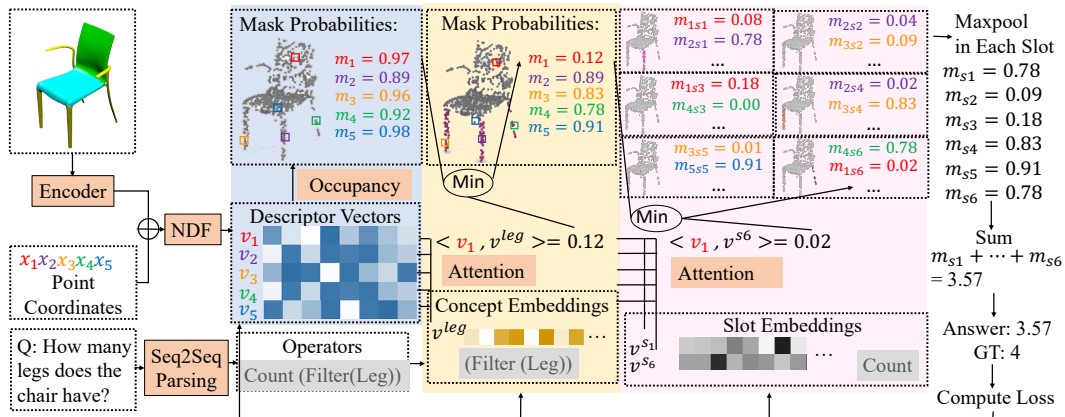

Figure 2: **Our 3D Concept Grounding (3D-CG) framework.** Given an input image and a question, 3D-CG first utilizes a Neural Descriptor Field (NDF) to extract a descriptor vector for each 3D point coordinate (here we take 5 coordinates with 5 different colors as examples), and a semantic parsing network to parse the question into operators. The mask probabilities of the coordinates are initialized with occupancy values. Concepts are also mapped to embedding vectors. Attention is calculated between the descriptors and concept embeddings to yield new mask probabilities and filter the set of coordinates being referred as the concept. We also define a count operator which segments a part category into different slots. The sum of the maxpooled mask probabilities of all slots can be used for counting, and the loss can be back propagated to optimize NDF as well as the embeddings. Semantic and instance segmentations emerge from question-answering supervision with 3D-CG.

design. Therefore, the entire framework can be end-to-end trained, and the gradient from the question answering loss can be utilized to optimize both the NDF perception module and the visual reasoning module and jointly learn all the embeddings and network parameters.

## 3.1 Model Details

### 3.1.1 Neural Descriptor Fields

In this paper, we utilize a neural field to map a 3D coordinate $\mathbf{x}$ using a descriptor function $f(\mathbf{x})$ and extract the feature descriptor of that 3D coordinate:

$$f(\mathbf{x}) : \mathbb{R}^3 \to \mathbb{R}^n, \quad \mathbf{x} \mapsto f(\mathbf{x}) = \mathbf{v} \tag{1}$$

where $\mathbf{v}$ is the descriptor representation which encodes information about shape properties (*e.g.*, geometry, appearance, *etc*).

In our setting, we condition the neural field on the partial point cloud $\mathcal{P}$ derived from RGB-D images. We use a PointNet-based encoder $\mathcal{E}$ to encode $\mathcal{P}$. The descriptor function then becomes $f(\mathbf{x}, \mathcal{E}(P)) = \mathbf{v}$. This continuous and differentiable representation maps each 3D coordinate $\mathbf{x}$ to a descriptor vector $\mathbf{v}$. Since the concepts to be grounded in most reasoning tasks focus on geometry and appearance, we leverage two pre-training tasks to parameterize $f$ and learn the features concerning these properties.

**Shape Reconstruction.** Inspired by recent works on using occupancy networks [29] to reconstruct shapes and learn shape representations, we also use an MLP decoder $\mathcal{D}_1$ which maps each descriptor vector $\mathbf{v}$ to an occupancy value: $\mathcal{D}_1 : \mathbf{v} \mapsto \mathbf{o} \in [0, 1]$, where the occupancy value indicates whether the 3D coordinate is at the surface of a shape.

**Color Reconstruction.** We use another MLP $\mathcal{D}_2$ to decode an RGB color of each coordinate: $\mathcal{D}_2 : \mathbf{v} \mapsto \mathbf{c} \in \mathbb{R}^3$, where the color value indicates the color of the 3D coordinate on the shape.

After learning decoders $D_i$ through these pre-training tasks, to construct the Neural Descriptor Field (NDF) $f(x)$, we concatenate all the intermediate activations of $D_i$ when decoding a 3D point $\mathbf{x}$ [41]. The resultant descriptor vector $\mathbf{v}$ can then be used for concept grounding. Since $\mathbf{v}$ consists of intermediate activations of $D_i$, they implicitly capture the hierarchical nature of a scene, with earlier activations corresponding to lower level features and later activations corresponding to higher level features.

### 3.1.2 Concept Quantization

Visual reasoning requires determining the attributes (*e.g.*, color, category) of a shape or shape part, where each attribute contains a set of visual concepts (*e.g.*, blue, leg). As shown in Figure 2, visual concepts such as legs, are represented as vectors in the embedding space of part categories. These concept vectors are also learned by our framework. To ground concepts in the neural fields, we calculate the dot products $\langle \cdot, \cdot \rangle$ between the concept vectors and the descriptor vectors from the neural field. For example, to calculate the score of a 3D coordinate belonging to the category `leg`, we take its descriptor vector $\mathbf{v}$ and compute $p = \langle \mathbf{v}, v^{leg} \rangle$, where $v^{leg}$ is the concept embedding of `leg`.

### 3.1.3 Semantic Parsing

To transform natural language questions into a set of primitive operators that can be executed on the neural field, a semantic parsing module similar to that in [53] is incorporated into our framework. This module utilizes a LSTM-based Seq2Seq to translate questions into a set of fundamental operators for reasoning. These operators takes concepts and attributes as their parameters (*e.g.*, `Filter(leg)`, `Query(color)`).

### 3.1.4 Neural operators on Neural Descriptor Fields

The operators extracted from natural language questions can then be executed on the neural field. Due to the differentiable nature of the neural field, the whole execution process is also fully-differentiable. We represent the intermediate results in a probabilistic manner: for the $i$-th sampled coordinate $\mathbf{x}_i$, it is represented by a descriptor vector $\mathbf{v}_i$ and has an attention mask $\mathbf{m}_i \in [0, 1]$. $\mathbf{m}_i$ denotes the probability that a coordinate belongs to a certain set, and is initialized using the occupancy value output by $\mathcal{D}_1$: $\mathbf{m}_i = \mathbf{o}_i$.

There are three kinds of operators output by the semantic parsing module, we will illustrate how each operator executes on the descriptor vectors to yield the outputs.

**Filter Operator.** The filter operator "selects outs" a set of coordinates belonging to the concept $c$ by outputting new mask probabilities:

$$\text{Filter}(c) : \mathbf{m}_i^c = min(\mathbf{m}_i, \langle \mathbf{v}_i, v^c \rangle) \tag{2}$$

**Query Operator.** The query operator asks about an attribute $a$ on selected coordinates and outputs concept $\hat{c}$ of the maximum probability:

$$\text{Query}(a) : \hat{c} = \arg\max min(\mathbf{m}_i, \langle \mathbf{v}_i, v^c \rangle), c \in a \tag{3}$$

**Count Operator.** The count operator intends to segment the coordinates in the same part category into instances and count the number of instances. Inspired by previous unsupervised segmentation methods which output different parts in various slots [26, 4], we also allocate a set of slots for different instances of a category. The $j$-th slot $s_j$ has its own embedding vector $v_j^s$. The score $s_{ij}$ of the $i$-th coordinate belonging to the $j$-th slot is also computed by dot product $s_{ij} = \langle \mathbf{v}, v^{s_j} \rangle$. We further take softmax to normalize the probabilities across slots. Since we want to count the instances of a part that was previously selected out, we also take the minimum of previous mask probabilities and the mask probabilities of each slot. The result of counting (*i.e.*, the number of part instances $n$) is obtained by summing up the maximum probability in each slot.

$$\text{Count}(c) : \mathbf{m}_{ij} = min(\mathbf{m}_i^c, \frac{s_{ij}}{\sum s_{i'j}}) \tag{4}$$

$$n = \sum_j \max_i \mathbf{m}_{ij} \tag{5}$$

### 3.1.5 Segmentation

Note that based on the above operators, not only visual reasoning can be performed, but we can also do semantic segmentation and instance segmentation.

**Semantic Segmentation.** We can achieve semantic segmentations by executing `Query(category)` on each coordinate and output the category $\hat{c}$ with the maximum probability for each point.

**Instance Segmentation.** We can perform instance segmentations by first doing semantic segmentation, and then for each output $\hat{c}$ we further execute $\texttt{count}(\hat{c})$.

## 3.2 Training Paradigm

**Optimization Objective.** During training, we jointly optimize the NDF and the concept embeddings by minimizing the loss as:

$$\mathcal{L} = \alpha \cdot \mathcal{L}_{\text{NDF}} + \beta \cdot \mathcal{L}_{\text{reasoning}} \tag{6}$$

Specifically, the NDF loss consists of two parts: the binary cross entropy classification loss between the ground-truth occupancy and the predicted occupancy, and the MSE loss between the ground-truth rgb value and the predicted rgb value.

We also define three kinds of losses for three types of questions: 1) For questions that ask about whether a part exist, we use an MSE loss $\|a - \max(\mathbf{m}_i)\|^2$ where $a$ is the ground-truth answer and $\max(\mathbf{m}_i)$ takes the maximum mask probability among all sampled 3D coordinates; 2) For questions that query about an attribute, we take the cross entropy classification loss between the predicted concept category $\hat{c}$ and the ground-truth concept category; 3) For counting questions, we first use an MSE loss $\|a - n\|^2$ between the answer and the number output by summing up the maxpool values of all the slots, and further add a loss to ensure that the mask probabilities of the top $K$ values in the top $a$ slots (*i.e.*, the slots with the maximum maxpool values) should be close to 1, where $K$ is a hyper-parameter. During training, we first train the NDF module only for $N_1$ epochs and then jointly optimize the NDF module and the concept embeddings. The Seq2seq model for semantic parsing is trained independently with supervision.

**Curriculum Learning.** Motivated by previous curriculum strategies for visual reasoning [28], we employ a curriculum learning strategy for training. We split the questions according to the length of operators they are parsed into. Therefore, we start with questions with only one neural operator (*e.g.*, "is there a yellow part of the chair" can be parsed into $\texttt{Filter(Color)}$).

# 4 Experiments

## 4.1 Experimental Setup

### 4.1.1 Dataset

Instead of object-based visual reasoning [19, 53, 28] where objects are spatially disentangled, which makes segmentations quite trivial, we seek to explore part-based visual reasoning where segmentations and reasoning are both harder. To this end, we collect a new dataset, $\texttt{PartNet-Reasoning}$, which focuses on visual question answering on the $\texttt{PartNet}$ [31] dataset. Specifically, we render approximately 3K RGB-D images from shapes of 4 categories: $\texttt{Chair}$, $\texttt{Table}$, $\texttt{Bag}$ and $\texttt{Cart}$, with 8 question-answer pairs on average for each shape. We have three question types: $\texttt{exist\_part}$ queries about whether a certain part exists by having the filter operator as the last operator; $\texttt{query\_part}$ uses query operator to query about an attribute (*e.g.*, part category or color) of a filtered part; $\texttt{count\_part}$ utilizes the count operator to count the number of instances of a filtered part. We are interested in whether the reasoning tasks can benefit the segmentations of the fine-grained parts, as well as whether the latent descriptor vectors by NDF can result in better visual reasoning.

### 4.1.2 Evaluation Tasks
**Reasoning.** We report the visual question answering accuracy on the $\texttt{PartNet-Reasoning}$ dataset w.r.t the three types of questions on all four categories.

**Segmentation.** We further evaluate both the performances of semantic segmentation and instance segmentation on our dataset. As stated in 3.1.5, semantic segmentation can be performed by querying the part category with the maximum mark probability of each coordinate, and filtering the coordinates according to category labels, and instance segmentation can be achieved by using the count operator on each part category. We report the mean per-label Intersection Over Union (IOU).

### 4.1.3 Baselines

We compare our approach to a set of different baselines, with details provided in the supp. material.

|  |  | PointNet+LSTM | MAC | NDF+LSTM | CVX+L | BAE+L | 3D-CG |
|---|---|---|---|---|---|---|---|
| Chair | exist_part | 52.3 | 65.2 | 55.7 | 71.9 | 72.4 | **85.4** |
|  | query_part | 41.6 | 53.9 | 54.2 | **72.3** | 70.5 | 68.7 |
|  | count_part | 63.4 | 78.1 | 71.6 | 48.8 | 68.0 | **92.2** |
| Table | exist_part | 55.1 | 66.4 | 61.3 | 68.5 | 71.0 | **80.3** |
|  | query_part | 43.6 | 51.2 | 54.4 | 69.7 | 73.6 | **75.1** |
|  | count_part | 35.5 | 55.3 | 50.1 | 30.7 | 45.7 | **90.9** |
| Bag | exist_part | 65.4 | 85.4 | 69.2 | 87.3 | 73.8 | **89.2** |
|  | query_part | 53.1 | 74.8 | 64.0 | **88.1** | 70.6 | 85.2 |
|  | count_part | 51.2 | 70.9 | **72.3** | 53.4 | 31.4 | 68.3 |
| Cart | exist_part | 49.7 | 75.3 | 61.7 | 79.1 | **91.0** | 90.1 |
|  | query_part | 50.1 | 64.0 | 57.0 | 72.3 | 81.5 | **86.3** |
|  | count_part | 41.6 | **82.1** | 67.3 | 46.6 | 47.3 | 74.8 |

Table 1: **Visual question answering accuracies.** `exist_part` queries about whether a certain part exists by having the filter operator as the last operator; `query_part` uses query operator to query about an attribute of a filtered part; `count_part` utilizes the count operator to count the number of instances of a filtered part. Point+LSTM, MAC and NDF+LSTM are methods based on neural networks. CVX+L and BAE+L are neural-symbolic methods which pre-segment the masks and ground concepts on the masks. 3D-CG outperforms baseline models by a large margin.

**Unsupervised Segmentation Approaches.** We compare 3D-CG with two additional 3D unsupervised segmentation models, and further consider how we may integrate such approaches with concept grounding. First, we consider **CVXNet** [7], which decomposes a solid object into a collection of convex polytope using neural implicit fields. In this baseline, a number of primitives are selected and a convex part is generated in each primitive. Next, we consider **BAENet** [4] which decomposes a shape into a set of pre-assigned branches, each specifying a portion of a whole shape. In practice, we tuned with the original codes of BAENet on our dataset and found that nearly all coordinates would be assigned to a single branch. This is probably because the original model is trained on voxel models, while ours is on partial point cloud and contains more complex shapes, thus posing more challenges. Therefore, for BAENet we use an additional loss which enforces that at least some of the branches should have multiple point coordinates assigned to positive occupancy value. The primitives in CVXNet and the branches in BAENet are similar to slots in our paper.

**Reasoning Baselines.** We compare our approach to a set of different reasoning baselines. First we consider **PointNet+LSTM** and **NDF+LSTM**, which use the features by PointNet or our NDF module, concatenated with the features of the questions encoded by a LSTM model. The final answer distribution is predicted with an MLP. Next, we compare with **MAC**, a commonly-used attention-based model for visual reasoning. We add a depth channel to the input to the model. Finally we compare with **CVX+L** and **BAE+L** are neural-symbolic models that use the similar language-mediated segmentation framework from [47]. Specifically, they first initiate the segmentations in the slots, and each slot has a slot feature vector. The operators from the questions are executed symbolically on the slot features. Question answering loss can be also propagated back to finetune the slot segmentations and features.

**Segmentation Baselines.** For the segmentation tasks, we compare our approach with unsupervised segmentation approaches **CVX** and **BAE** described above. We further construct language-mediated variants **CVX-L** and **BAE-L**. For semantic segmentation by CVX and BAE, we use the manual grouping methods as in [7]. For semantic segmentation by language-mediated models, we use the filter operator to filter the slots belonging to the part categories.

## 4.2 Experimental Results

### 4.2.1 Reasoning & Concept Grounding

Table 1 shows the VQA accuracy among all models on our dataset. We can see that overall, our 3D-CG outperforms baseline modeles by a large margin. Language-mediated neural-symbolic methods (CVX+L & BAE+L) are better than neural methods in the `exist_part` and `query_part` question types, but in general they are far worse than our method. This is because the slot-based representations pre-select the parts and make it easier for the concepts to attend to specific regions. However, wrong pre-segmentations are also hard to be corrected during the training of reasoning.

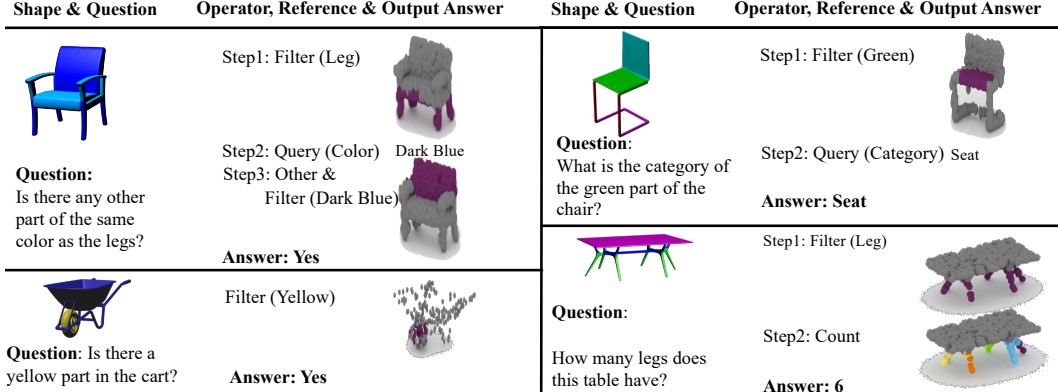

| Shape & Question | Operator, Reference & Output Answer | Shape & Question | Operator, Reference & Output Answer |
|---|---|---|---|

Step1: Filter (Leg)

Step2: Query (Color) Dark Blue
Step3: Other &
  Filter (Dark Blue)

**Question:**
Is there any other part of the same color as the legs?

**Answer: Yes**

Step1: Filter (Green)

Step2: Query (Category) Seat

**Question:**
What is the category of the green part of the chair?

**Answer: Seat**

Filter (Yellow)

**Question:** Is there a yellow part in the cart?

**Answer: Yes**

Step1: Filter (Leg)

Step2: Count

**Question:**

How many legs does this table have?

**Answer: 6**

Figure 3: **Qualitative Illustration.** Examples of the reasoning process of our proposed 3D-CG. Given an input image of a shape and a question, we first parse the question into steps of operators. We visualize the set of points being referred to by the operator via highlighting the regions where mask probabilities > 0.5. As is shown,our 3D-CG can make reference to the right set of coordinates, thus correctly answering the questions.

The accuracies for the `count_part` type further demonstrates this point, where language-mediated baselines have extremely low accuracies, and segmentations from Figure 4a also show that even with supervision from question answering loss, BAE and CVX cannot segment the instances of a part category (*e.g.*, legs and wheels). Neural methods such as MAC and NDF+LSTM perform well in the `count_part` type in some categories, probably due to some shortcuts of the `PartNet` dataset (*e.g.*, most bags have two shoulder straps). In comparison to both neural and neural-symbolic methods, our method performs well in all three question types. This is because the regions attended by concepts are dynamically improved with the question answering loss, while the advantage of explicit disentanglement of concept learning and reasoning process is maintained.

Figure 3 shows some qualitative examples of the reasoning process of our method. Specifically, the questions are parsed into a set of neural operators. For each step, the neural operator takes the output of the last step as its input. For the filter operator, we visualize the attended regions with mask probabilities greater than 0.5. From the figure, we can see that our method can attend to the correct region to answer the questions, as well as segment the instances correctly for counting problems. This attention-based reasoning pipeline is also closer than previous neural-symbolic methods to the way that humans perform reasoning. For example, when asked the question "What is the category of the green part of the chair?", humans would directly pay attention to the green region regardless of the segmentations of the rest of the chair. However, previous neural symbolic methods [53, 28, 47] segment the chair into different parts first and then select the green part, which is very unnatural.

### 4.2.2 Segmentation

Table 2 shows the semantic segmentation and instance segmentation results, and Figure 4a visualizes some qualitative examples. We can see that overall, our 3D-CG can better segment parts than unsupervised or language-mediated methods. Other methods experience some common problems such as failing to segment the instances within a part category (*e.g.*, one of the legs is always merged with the seat), or experience unclear boundaries and segment one part into multiple parts (*e.g.*, segmenting the chair back or the tabletop into two parts). In general, language-mediated methods are better than pure unsupervised methods, which indicates that the supervision of question answering does benefit the segmentation modules. However, a lot of the wrong segmentations cannot be corrected by reasoning, resulting in bad results for counting questions for these models. In contrast, our 3D-CG can learn segmentations well using question answering loss, mainly because the segmentations emerge from question-answering supervision without any restrictions.

For fair comparison with baselines, we first control the reconstruction part and use the ground-truth voxels for the segmentation evaluations only. We also provide additional experimental results when the segmentations are performed on the reconstructed voxels, which is shown in Table 3.

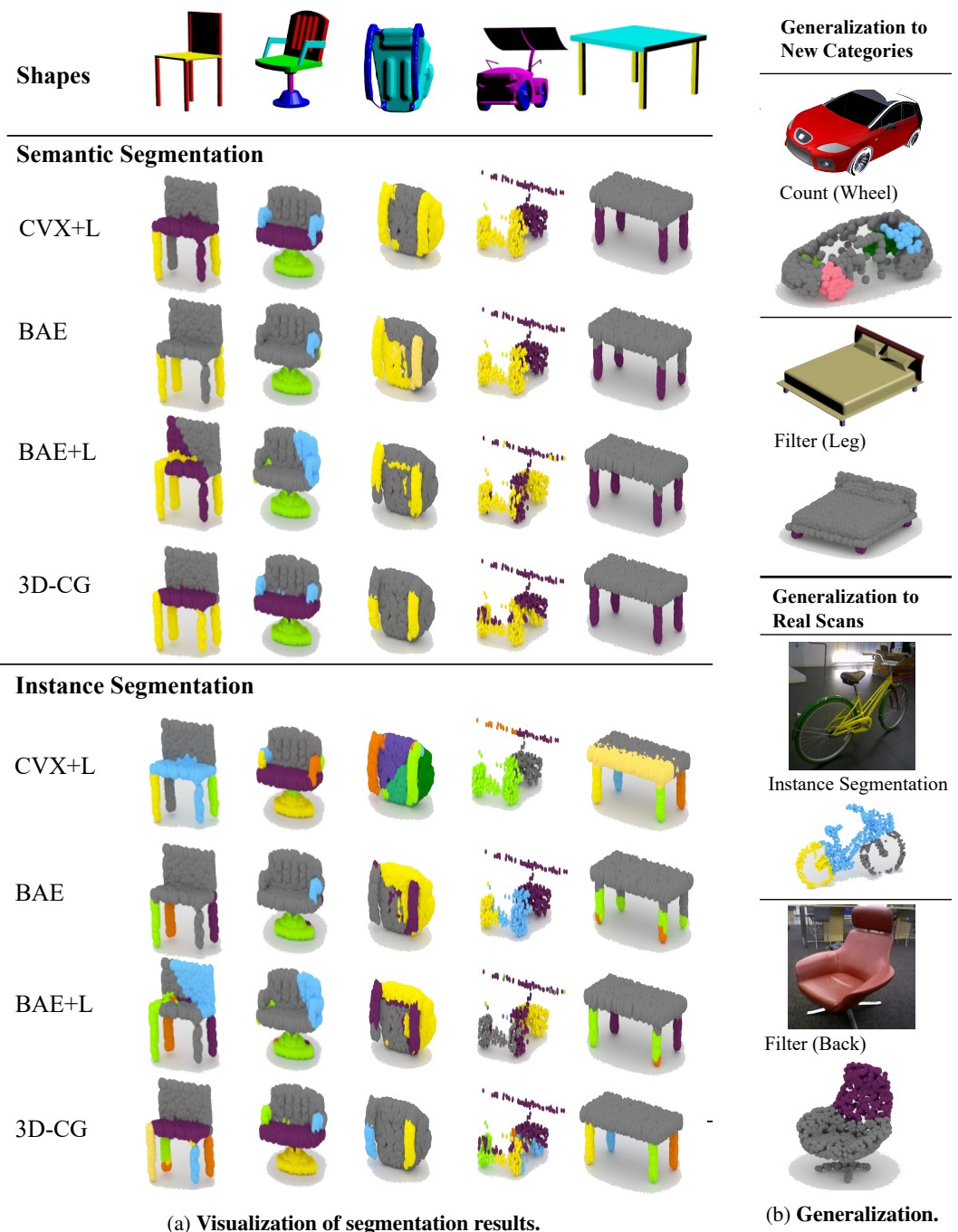

(a) **Visualization of segmentation results.**

(b) **Generalization.**

Figure 4: Visualization of segmentation as well as generalization. Here we use ground-truth voxel values for better visualization. CVX and BAE are unsupervised segmentaton methods. CVX+L and BAE+L are language-mediated methods which utilize question answering loss to finetune the segmentation module. 3D-CG has better performances in both semantic and instance segmentations, while other methods suffer from merging parts or segmenting a part into multiple parts. It can also generalize well to unseen categories and real scans.

### 4.2.3    Generalization

In Figure 4b, we show some qualitative examples of how our 3D-CG trained on seen categories can be generalized directly to unseen categories and real scans. We show results of semantic and instance segmentations, as well as visualize the parts that are referred to in the questions.

| | Chair | Table | Bag | Cart | Mean | | Chair | Table | Bag | Cart | Mean |
|---|---|---|---|---|---|---|---|---|---|---|---|
| CVX | 62.3 | 74.2 | 66.0 | 44.2 | 61.7 | CVX | 44.1 | 40.6 | 31.2 | 22.5 | 34.6 |
| CVX+L | 64.6 | 74.5 | **70.2** | 51.3 | 65.2 | CVX+L | 42.6 | 51.2 | 41.3 | 20.4 | 38.9 |
| BAE | 49.5 | 71.0 | 64.1 | 49.7 | 58.6 | BAE | 53.3 | 32.2 | 39.8 | 34.0 | 39.9 |
| BAE+L | 56.3 | 72.3 | 69.8 | 46.9 | 61.3 | BAE+L | 50.1 | 47.0 | 34.1 | 36.6 | 42.0 |
| 3D-CG | **76.6** | **79.3** | 67.3 | **54.2** | **69.4** | 3D-CG | **68.5** | **71.2** | **47.2** | **40.5** | **56.9** |

(a) Semantic Segmentation IOU  (b) Instance Segmentation IOU

Table 2: **Segmentation Results on Ground-Truth Occupancy Values**. 3D-CG outperforms all unsupervised / language-mediated baseline models.

| | Chair | Table | Bag | Cart | Mean | | Chair | Table | Bag | Cart | Mean |
|---|---|---|---|---|---|---|---|---|---|---|---|
| CVX | 38.1 | 49.5 | 40.3 | 26.0 | 38.5 | CVX | 29.4 | 26.4 | 18.2 | 13.4 | 21.9 |
| CVX+L | 34.0 | 39.0 | 38.3 | 29.2 | 35.1 | CVX+L | 22.9 | 28.2 | 22.6 | 13.8 | 21.9 |
| BAE | 36.3 | 50.1 | 50.0 | 38.5 | 43.8 | BAE | 38.7 | 22.7 | 28.1 | 24.8 | 28.6 |
| BAE+L | 41.1 | 51.6 | 51.7 | 34.5 | 44.8 | BAE+L | 37.3 | 34.5 | 25.1 | 26.7 | 30.9 |
| 3D-CG | **69.3** | **65.7** | **56.9** | **48.2** | **60.0** | 3D-CG | **57.1** | **58.7** | **42.0** | **35.4** | **48.3** |

(a) Semantic Segmentation IOU  (b) Instance Segmentation IOU

Table 3: **Segmentation Results on Preidcted Occupancy Values**.

For generalizing to new categories, we use the model trained on carts to ground and count the instances of the concept "wheel" in cars. We can see that the instance segmentation results on the wheels are not perfect because one wheel is in wrong position. However, most of the wheels are detected and the model manages to output the right count. We also use the model trained on chairs to filter the legs of a bed. All the legs are successfully selected out by our 3D-CG.

We also use real 3D scans from the RedWood dataset [6] to estimate 3D-CG's ability to generalize to real scenes. We use a single-view scan to reconstruct partial point cloud and remove the background such as the floor. We input the partial point cloud into our 3D-CG and output the segmentations on the ground-truth voxels. For generalizing to bicycles, we use the 3D-CG model trained on carts. We can see that 3D-CG can find all instances in the bicycle and detect both wheels. Furthermore, 3D-CG trained on chairs can also be generalized to chairs in real scans.

## 5   Discussion

**Conclusion.**   In this paper, we propose 3D-CG, which leverages the continuous and differentiable nature of neural descriptor fields to segment and learn concepts in the 3D space. We define a set of neural operators on top of the neural field, with which not only can semantic and instance segmentations emerge from question-answering supervision, but visual reasoning can also be well performed.

**Limitations and Future Work.**   A limitation of our underlying framework with 3D-CG is that while we show transfer results on real scenes, our approach is only trained with synthetic scenes and questions. While we believe our proposed operations is general-purpose, an interesting direction of future work would be scaling our framework to directly train on complex real-world scenes, and ground more concepts from natural language on these real scenes. A promising direction of future work would be to explore the combination of large pre-trained visual-language models and volumetric rendering on multi-view images.

## Acknowledgments and Disclosure of Funding

This work was supported by MIT-IBM Watson AI Lab and its member company Nexplore, Amazon Research Award, ONR MURI, DARPA Machine Common Sense program, ONR (N00014-18-1-2847), and Mitsubishi Electric.

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
