# Supplementary Material for 3D Concept Grounding on Neural Fields

**Yining Hong**
University of California, Los Angeles

**Yilun Du**
Massachusetts Institute of Technology

**Chunru Lin**
Shanghai Jiao Tong University

**Joshua B. Tenenbaum**
MIT BCS, CBMM, CSAIL

**Chuang Gan**
UMass Amherst and MIT-IBM Watson AI Lab

## Overview

In this appendix, we supplement our paper by providing more qualitative examples and details about our model to help readers better understand our paper.

This appendix is organized as follows.

- In Sec. A, we provide more qualitative examples about how 3D-CG performs reasoning steps as well as how our model generalizes to real-world dataset.
- In Sec. B, we provide more details about our 3D-CG model, including the detailed architecture of NDF, the semantic parsing module and the curriculum learning strategies.
- In Sec. C, we provide more details about the baselines models.
- In Sec. D, we provide quantitative results of generalization to novel categories.
- In Sec. E, we discuss the potential societal impacts of this paper.

36th Conference on Neural Information Processing Systems (NeurIPS 2022).

# A  More Qualitative Examples

## A.1  QA Examples

In Figure 1, we show more examples of the reasoning process of 3D-CG. We can see that overall, our model can make reference to the correct region to answer the questions, although sometimes the segmentation might be a little noisy (*e.g.,* for the bag a part of the shoulder strap is not highlighted, and the highlighted part of cart contains part of the wheel.). Also, for counting problems, we can see that our model can segment the legs and make the right prediction. However, for the chair with five legs, although our model can assign five colors to five legs, some of the legs are overlapped with each other by colors.

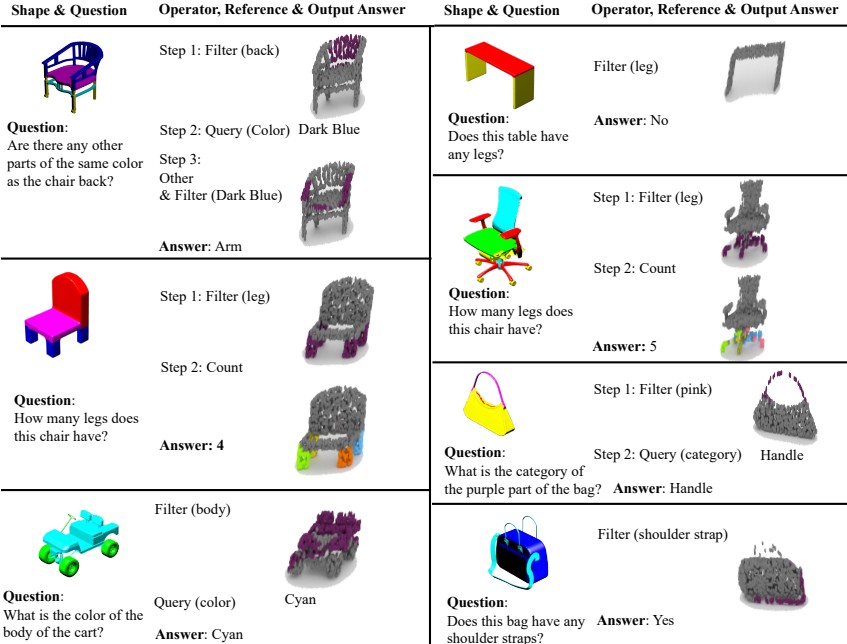

Figure 1: More qualitative examples that illustrate the reasoning process of 3D-CG. Given an input image of a shape and a question, we first parse the question into steps of operators. We visualize the set of points being referred to by the operator via highlighting the regions where mask probabilities > 0.5. As is shown, our 3D-CG can make reference to the right set of coordinates, thus correctly answering the questions.

## A.2  Generalization

Figure 2 shows more examples on the RedWood dataset. As can be seen, the model trained on PartNet can be easily generalized to real-world scans.

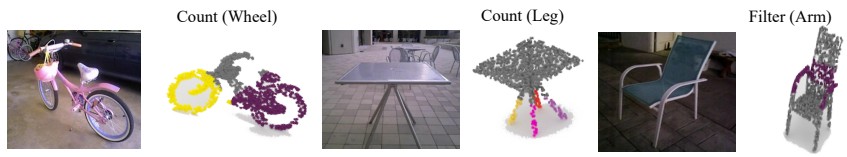

Figure 2: More qualitative examples of how our 3D-CG generalizes to the RedWood dataset.

# B  Details on 3D-CG

## B.1  NDF

We first train the reconstruction task solely for 50,000 iterations.

For encoder, we use a PointNet-based encoder network with ResNet blocks. We use 5 resnet blocks, each with input dim 256 and output dim 128. At each step the output is concatenated with the input to produce the input of the next step. To enable communication between points at lower layers, we also add pooling and expansion layers between the ResNet-blocks. After the ResNet-blocks, the final output is pooled using max-pooling and then projected to a 128 embedding vector using a fully-connected layer.

For the decoder, we first concatenate the coordinates and the point cloud latent feature vector. The new input is passed through 5-layer resnet blocks. All the resnet blocks have a hidden size of 128. The features $\mathbf{v}$ of the blocks are concatenated together and passed through a fully-connected layer to produce the 1-dimensional occupancy value. For color reconstruction, the features go through 3 linear layers with hidden size 128 and in the last layer, produce 3-dimensional r,g,b value. The batch size is 4. The learning rate is 1e-4. All embeddings have the dim 2049.

To learn the parameters of the NDF, we randomly sample $N = 3,000$ points in the 3D bounding volume of each object under consideration. For a partial point cloud $P$, the NDF loss is:

$$\mathcal{L}_{NDF} = \sum_{j=1}^{N}\{-[\mathbf{o}_j \log \mathcal{D}_1(f(\mathbf{x_j}, \mathcal{E}(P))) + (1 - \mathbf{o}_j)\log(1 - \mathcal{D}_1(f(\mathbf{x_j}, \mathcal{E}(P))))]$$
$$+ \|\mathcal{D}_2(f(\mathbf{x_j}, \mathcal{E}(P))) - \mathbf{c}_j\|_2^2\}$$

## B.2  Semantic Parsing

Our semantic parser is an attention-based sequence to sequence (seq2seq) model with an encoder-decoder structure similar to that in [2]. The encoder is a bidirectional LSTM [1]. The decoder is a similar LSTM that generates a vector from the previous token of the output sequence. Both the encoder and decoder have two hidden layers with a 256-dim hidden vector. We set the dimensions of both the encoder and decoder word vectors to be 300.

## B.3  Curriculum learning

In Figure 3, we show detailed strategies of curriculum training for 3D-CG. During the whole training process, we gradually add more visual concepts and more complex question examples into the model, which is shown in Figure 3. In general, the whole training process is split into 3 stages. First, we learn `color` concepts. We want to learn individual concepts like `red` or `seat`. However, since most chairs contain certain parts like `back` and `seat`, we therefore learn the `color` concepts first by asking questions like "Is there a red part in the chair?" For the second lesson, after learning about `color` concepts like in the figure, we can then attend to the red region, and we can ground different part categories by asking questions like "What is the category of the red part?". Finally, we can train more complex questions altogether. Each lesson is trained for approximately 50,000 iterations.

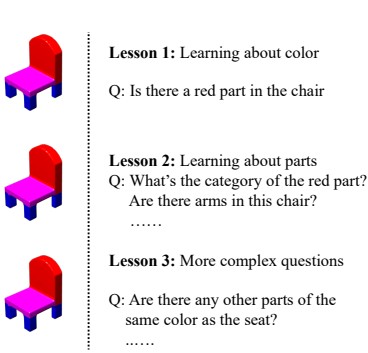

**Lesson 1:** Learning about color

Q: Is there a red part in the chair

**Lesson 2:** Learning about parts
Q: What's the category of the red part?
Are there arms in this chair?
......

**Lesson 3:** More complex questions

Q: Are there any other parts of the same color as the seat?
......

Figure 3: The curriculum learning strategy of 3D-CG.

# C  Datails on Baseline Models

## C.1  Details on Language-mediated models

We explain in detail how CVX-L and BAE-L work for segmentation and reasoning, as is shown for
Figure 4. As the original CVX/BAE model, we use the same NDF as in 3D-CG, except that instead of
decoding the descriptor vectors into N*1 occupancy values, we first decode them into N*S occupancy
values for different slots, and then maxpool to get the overall occupancy values for reconstruction.
For reasoning, the descriptor vectors of each slot is mean pooled to get a part-centric representation.
The question also goes through a semantic parsing module like 3D-CG, and attention is calculated
between the concept embedding vector and the descriptor vector of each slot to identify whether the
part in one slot can be accounted for a concept. The reasoning loss can also be back-propagated to the
NDF module and the concept embeddings. We use the same curriculum learning strategy as 3D-CG
and train for the same number of epochs.

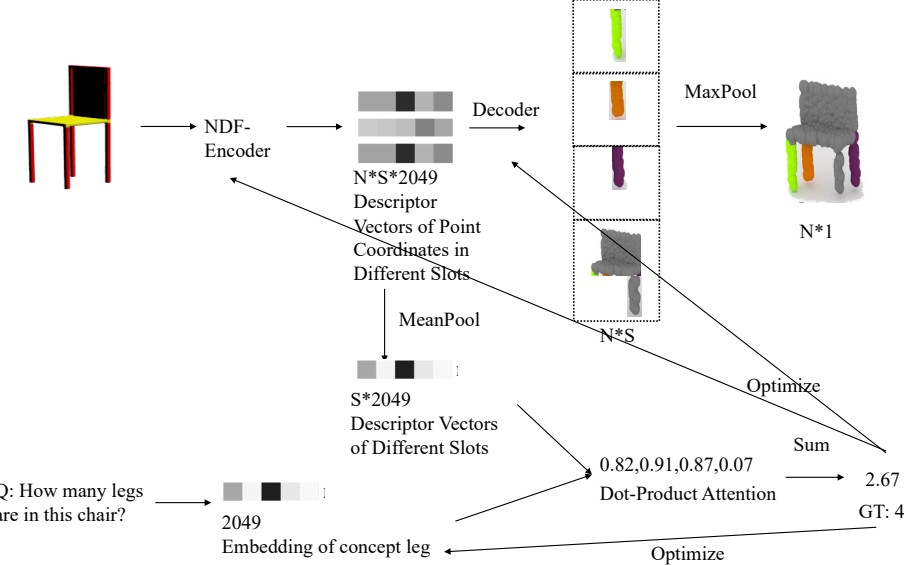

Figure 4: Detailed framework of CVX-L / BAE-L, where N denotes the number of sampled points, and S denotes
the number of slots.

## C.2  Neural-network based baselines

For MAC, we use an ImageNet-pretrained ResNet-101 to extract $14 \times 14 \times 1024$ feature maps for
MAC. For PointNet+LSTM, we use a 2048 dimensional feature from the last layer. For NDF+LSTM,
we mean-pool the descriptor vectors of all point coordinates to get 2049-dimensional vectors. All
models are trained for 50 epochs.

# D  Quantitative Results on Generalization to Novel Categories

In this section, we provide the quantitative performances on novel categories for both baselines and 3D-CG.

|     |            | PointNet+LSTM | MAC  | NDF+LSTM | CVX+L | BAE+L | 3D-CG |
|-----|------------|---------------|------|----------|-------|-------|-------|
| Bed | exist_part | 46.6          | 63.7 | 61.5     | 63.4  | 58.6  | **75.8** |
|     | query_part | 32.1          | 49.8 | 46.7     | 50.0  | 51.6  | **59.1** |
|     | count_part | 45.2          | 90.6 | 85.9     | 55.3  | 49.5  | **98.7** |
| Car | exist_part | 40.7          | 67.8 | 59.7     | 54.3  | **69.4** | 64.8 |
|     | query_part | 42.5          | 51.5 | 52.6     | 56.3  | **70.1** | 68.8 |
|     | count_part | 34.3          | 58.9 | 53.4     | 28.6  | 21.5  | **85.7** |

Table 1: **quantitative performances on novel categories.**

# E   Societal Impacts

Our work is of broad interest to the computer vision and machine learning communities. Our proposed no inherent ethical or societal issues on its own, but inherit those typical of learning methods, such as capturing the implicit biases of data. Our work aims to enable more interpretable reasoning, and may serve as an inspiration for works that aims to perform less black-box reasoning with gradient-based learning.