# OpenReview forum: "3D Concept Grounding on Neural Fields"
_NeurIPS.cc/2022/Conference — NeurIPS 2022 Accept_

### Official Review · Reviewer_gaS1 · 2022-07-11

**Rating:** 6
**Confidence:** 4
**Soundness:** 3 good
**Presentation:** 4 excellent
**Contribution:** 3 good

**Summary:**

The paper proposes a weakly-supervised method that learns to ground 3D concepts with RGBD inputs. The key idea is to introduce Neural Descriptor Field (NDF) to neural operators (filter, query, and counting). 3D concepts are shown to emerge from training on VQA dataset. They extend their method to part semantic segmentation and part instance segmentation and show some genarlziation ability to unseen classes and real scans.


**Questions:**

see weakness 2

**Limitations:**

The authors have addressed the limitations and potential negative societal impact of their work

**Strengths And Weaknesses:**

Strength
+ The paper aims to lift concept reasoning to 3D space. I think the biggest strength is the weakly-supervised method. The method leverages differentiable neural operators such that 3D grounding can be trained with question answer pairs. It also naturally allows for better interpretability.
+ With a bunch of modules, the paper is very well written and easy to follow. Toy examples are nicely introduced to help readers understand.
+ Part semantic segmentation and instance segmentation are also shown as special cases of the proposed method.
+ The results look convincing to me. Since the problem is quite novel, the authors do a good job to compare with baselines.

Weakness
1. The proposed method focuses on object-centric reasoning and the method may be hard to scale up to more complicated scenes because the NDF considers the whole 3D space as a continuous, per-point query space. However, Table1 may suggest that some primitive-based methods may have advantages if their emerging part aligns with the language guide. -- It is more of my personal suggestion than weakness.
2. The generalization claim is quite weak as only some qualitative results are shown. For example, I wonder about the quantitative performance on novel categories for both baselines and their method.

---

> ### Author Response · Authors · 2022-08-02
> **Response to Reviewer gaS1**
>
> *Thank you for your insightful and constructive comments! We have added additional experiments and modified our paper according to your comments.*
>
> &nbsp;&nbsp;
>
> > **The proposed method focuses on object-centric reasoning and the method may be hard to scale up to more complicated scenes because the NDF considers the whole 3D space as a continuous, per-point query space.**
>
> &nbsp;&nbsp;&nbsp; Our framework is a general framework that can be applied to arbitrary neural fields. We show that our framework can easily scale up to more complicated scenes by providing qualitative examples on real-world scenes, which are shown on the demo website (https://sites.google.com/view/3d-cg/home). Here, we briefly illustrate how to extend the framework to more general-purpose visual reasoning tasks:
>
> 1) Same as our paper, we can get feature descriptors from the neural descriptor field (NDF). Since 3D occupancy values are hard to get for real-world scenes, we can train the NDF to volumetrically render the features for the multiview images.
> 2) To increase the diversity of natural language questions, we can use large pre-trained models such as CLIP [1] to initialize the concept embeddings.
> 3) The resultant 3D descriptor field and concept embeddings can be directly integrated with our proposed operators.
>
>
> > **The generalization claim is quite weak as only some qualitative results are shown. For example, I wonder about the quantitative performance on novel categories for both baselines and their method.**
>
> &nbsp;&nbsp;&nbsp; Thank you for pointing it out! We have added the quantitative results to Section E in the Supplementary Material, which are also shown below.
>
> Table C. Quantitative results of generalization
>
> |          | | PointNet+LSTM | MAC | NDF+LSTM | CVX+L | BAE+L | 3D-CG |
> | -------- | --------| ---- | ---- | ---- | ---- | ---- | ---- |
> | Bed | exist_part | 46.6 | 63.7 | 61.5 | 63.4 | 58.6 | 75.8 |
> |  | query_part  | 32.1 | 49.8 | 46.7 | 50.0 | 51.6 | 59.1 |
> |  | count_part | 45.2 | 90.6 | 85.9 | 55.3 | 49.5 | 98.7 |
> | Car | exist_part | 40.7 | 67.8 | 59.7 | 54.3 | 69.4 | 64.8 |
> |  | query_part | 42.5 | 51.5 | 52.6 | 56.3 | 70.1 | 68.8 |
> |  | count_part | 34.3 | 58.9 | 53.4 | 28.6 | 21.5 | 85.7 |
>
> &nbsp;&nbsp;
>
> [1] Learning Transferable Visual Models From Natural Language Supervision. Alec Radford et al.
>
> &nbsp;&nbsp;
>
>
> *We sincerely appreciate your comments. Please feel free to let us know if you have further questions. Thank you!*
>
> Best,
> Authors

---

### Official Review · Reviewer_Xamz · 2022-07-11

**Rating:** 6
**Confidence:** 4
**Soundness:** 3 good
**Presentation:** 4 excellent
**Contribution:** 4 excellent

**Summary:**

This paper studies the problem of the approach and the effectiveness of the 3D concept grounding using neural implicit field representation. The authors focus on several types of questions and develop a pipeline including multiple stages. They also create a dataset containing the necessary types of questions to assist the experiment. The results show that by using 3D concept grounding information, the trained models outperform previous works and generalize well to unseen shapes.

**Questions:**

Please see my comments in the "weaknesses" part.

**Limitations:**

The limitation is addressed in the paper.

**Strengths And Weaknesses:**

Pros:
+ The paper is well written.
+ This paper studies a novel and interesting problem.
+ A dataset contains 3K images and 3 types of questions are created to facilitate the experiments.
+ A novel framework based on the neural descriptor field with multiple stages and attention blocks is proposed.
+ The results that demonstrate superior performance and generalization are promising.

Cons:
- The experimenting dataset is relatively small.
- The framework seems to be designed for a specific purpose (i.e. answering limited types of questions) rather than a general-purpose one. I don't see this framework can be easily extendable.
- A concern would be that it is well known that 3D labeling/datasets are hard to obtain, not to mention combined with question-answering data.

---

> ### Author Response · Authors · 2022-08-02
> **Response to Reviewer Xamz**
>
> *We appreciate the positive and insightful comments from you! We address your concerns in details below.*
>
> &nbsp;&nbsp;
>
> > **Q1 & 2: The experimenting dataset is relatively small. The framework seems to be designed for a specific purpose (i.e. answering limited types of questions) rather than a general-purpose one. I don't see this framework can be easily extendable.**
>
>
> &nbsp;&nbsp;&nbsp;&nbsp; Our framework is a general framework that can be applied to abitrary neural fields and visual reasoning tasks. We show that our framework can be easily extendable by providing qualitative examples on real-world scenes, which are shown on the demo website (https://sites.google.com/view/3d-cg/home). Here, we briefly illustrate how to extend the framework to more general-purpose visual reasoning tasks:
>
> 1) Same as our paper, we can get feature descriptors from the neural descriptor field (NDF). Since 3D occupancy values are hard to get for real-world scenes, we can train the NDF to volumetrically render the features for the multiview images.
> 2) To increase the diversity of natural language questions, we can use large pre-trained model such as CLIP [1] to initialize the concept embeddings.
> 3) The resultant 3D descriptor field and concept embeddings can be directly integrated with our proposed operators.
>
> > **Q3: A concern would be that it is well known that 3D labeling/datasets are hard to obtain, not to mention combined with question-answering data.**
>
> * **3D-CG may release the burden on 3D labelling.**
>
>     We agree that obtaining fine-grained labeling 3D dataset is very challenging. Generally, there are two kinds of approaches to achieve fine-grained 3d lableing. The first is to manullay label 3D segmentations, which can be really time-consuming and labor-intensive when it comes to real-world scenarios (*e.g.,* self-driving). Instead, our framework can naturally segment and ground the concepts on the 3D scenes with the supervision of question answering. In general, question-answer pairs are much easier to annotate than segmentations or bounding boxes. One possible way to do this is to automatically generate some questions using templates [2], and ask humans to answer these questions. The concepts in the question-answer pairs might be naturally grounded in the 3D representations and segment the scenes. In this way, a promising direction may be utilizing our framework for **human-in-the-loop auto-labelling** for 3D scenes.
> * **Our framework is flexible with different neural field backbones that require less 3D supervision**.
>
>     Instead of occupancy-based neutral fields which require 3D data, We may use neural field architectures like neural radiance field which infers the 3D representations directly from **multi-view images**. Our results on the demo website show this promising direction, which might further release the burden on 3D data.
>
> &nbsp;
>
> [1] Learning Transferable Visual Models From Natural Language Supervision. Alec Radford et al.
>
> [2] CLEVR: A Diagnostic Dataset for Compositional Language and Elementary Visual Reasoning. Justin Johnson et al.
>
> &nbsp;
>
> *We hope that the provided new experiments and additional explanations have convinced you of the merits of our work. Please do not hesitate to contact us if you have other concerns.*
>
> *We appreciate your time! Thank you so much!*
>
> Best,
> Authors

---

### Official Review · Reviewer_1ERZ · 2022-07-11

**Rating:** 8
**Confidence:** 4
**Soundness:** 3 good
**Presentation:** 3 good
**Contribution:** 4 excellent

**Summary:**

This paper presents a method for grounding natural language concepts in 3D neural fields of shape and appearance. The main contribution is to use neural descriptor fields, a continuous field and ground it in language concepts. The input to the method is a set of shapes/appearance and QA about that shape. No other supervision is used. Specifically, the paper looks at binary questions, counting questions, and segmentation questions. Furthermore, the paper presents a slot attention module and neural operators for counting/segmentation.

**Questions:**

In the rebuttal, it would be nice to see a plan to include more details that are currently missing in the paper (see above).

Will the PartNet-Reasoning dataset be publicly released?

**Limitations:**

Yes, there is a discussion of limitations and societal impact. This discussion is adequate.

**Strengths And Weaknesses:**

## Strengths

- I quite like the idea of grounding concepts to neural fields rather than to 3D shape/appearance. This makes it continuous and offers previously unseen differentiability properties.
- The paper grounds concepts to descriptor fields without any explicit supervision. This is an exciting direction that could open up new research directions.
- The proposed neural operators, particularly for counting is novel to my knowledge.
- The results shown are promising and show improvement compared to baselines. I especially appreciated the inclusion of real image results.

## Weaknesses

- Firstly, it's not very clear what class of data the method operates on. Is it scenes or object classes? Only in the experiment section is this made clear (PartNet-Reasoning). This is an important detail that the paper ought to mention earlier.

- The paper seems to introduce a depth-map conditioned neural field (line 123), but details and differences to conventional neural shape fields are missing.

- Generally, the paper is lacking many details, especially in the shape and color representation parts, and the language parsing part. For instance, what kind of parsing does the seq2seq model do? Why were these three operators chosen? In the loss description, how exactly is L_{NDF} computed? Is there a reconstruction module that is missing in Figure 2.

- In terms of related work, the paper could have done a better job summarizing the latest advances. I am afraid that the related work is limited to a small set of works. I would recommend the authors take a look at a couple of recent works and papers cited within.

*Geometry Processing with Neural Fields, Yang et al. 2021

*Neural Fields in Visual Computing and Beyond, Xie et al. 2022

*Advances in Neural Rendering, Tewari et al. 2022

- In the datasets, how are the ~8 questions+answers per shape selected? Are these all related to the categories of tasks that the paper intends to solve, or are they more general? Also, the dataset has *both* questions and answers, not just questions. This should be mentioned in line 214.

Overall, this is really nice work that I would like to see at NeurIPS. The paper opens up new research directions that move the needle both in terms of 3D shape representations, learning with fewer examples, and concept grounding to 3D shapes. However, the related work section needs to be more comprehensive.

---

> ### Author Response · Authors · 2022-08-02
> **Response to Reviewer 1ERZ (Part 2/2)**
>
> > **Q3.3:  In the loss description, how exactly is L_{NDF} computed?**
>
> * We explained this in Line 205-207 in the paper: "the NDF loss consists of two parts: the binary cross entropy classification loss between the ground-truth occupancy and the predicted occupancy, and the MSE loss between the ground-truth rgb value and the predicted rgb value."
> * We further add one equation in the supplementary material Line 39:
>
>   $ \mathcal{L}=\sum_{j=1}^{N} - [\mathbf{o}_{j} \log \mathcal{D}_1(f(\mathbf{x_j}, \mathcal{E}(P))) +  (1 - \mathbf{o}_j) \log (1 - \mathcal{D}_1(f(\mathbf{x_j}, \mathcal{E}(P))))] + \|\mathcal{D}_2(f(\mathbf{x_j}, \mathcal{E}(P)))-\mathbf{c}_j\|_2^2$
>
> > **Q3.4: Is there a reconstruction module that is missing in Figure 2**
>
> &nbsp; &nbsp; The reconstruction module is shown in the blue column in Figure 2, with the *Occupancy* arrow pointing to the reconstructed shape.
>
> > **Q4: In terms of related work, the paper could have done a better job summarizing the latest advances. I am afraid that the related work is limited to a small set of works. I would recommend the authors take a look at a couple of recent works and papers cited within.**
>
> &nbsp; &nbsp; Thank you for pointing out the related works! In our revised manuscript, we have added a lot more related works (Line 94-107), especially those related to our paper.
>
> > **Q5.1: Are these all related to the categories of tasks that the paper intends to solve, or are they more general?**
>
> &nbsp; &nbsp; They are more related to the categories of tasks that the paper intends to solve. As illustrated in the response to Q3.2, these questions focus on the essential visual reasoning abilities that are also explored in previous works [1,2,3,4].
>
> > **Q5.2: Also, the dataset has both questions and answers, not just questions. This should be mentioned in line 214.**
>
> &nbsp; &nbsp; Please see Line 231 of the revised version.
>
> > **Q6: Will the PartNet-Reasoning dataset be publicly released?**
>
> &nbsp; &nbsp; Yes! The dataset, codes and checkpoints will be publicly available upon acceptance.
>
> &nbsp; &nbsp;
>
> [1] CLEVR: A Diagnostic Dataset for Compositional Language and Elementary Visual Reasoning. Justin Johnson, B. Hariharan, L. V. D. Maaten, Li Fei-Fei, C. L. Zitnick, Ross B. Girshick. CVPR 2017.
>
> [2] CLEVRER: CoLlision Events for Video REpresentation and Reasoning. Kexin Yi*, Chuang Gan*, Yunzhu Li, Pushmeet Kohli, Jiajun Wu, Antonio Torralba, Joshua B. Tenenbaum . ICLR 2020.
>
> [3] Neural-Symbolic VQA: Disentangling Reasoning from Vision and Language Understanding. Kexin Yi*, Jiajun Wu*, Chuang Gan, Antonio Torralba, Pushmeet Kohli, Joshua B. Tenenbaum. NeurIPS2018.
>
> [4] The Neuro-Symbolic Concept Learner: Interpreting Scenes, Words, and Sentences From Natural Supervision.  Jiayuan Mao, Chuang Gan, Pushmeet Kohli, Joshua B. Tenenbaum, Jiajun Wu. ICLR2019
>
> &nbsp; &nbsp;
>
> *Please let us know if you have any further questions for our paper. We really appreciate your time! Thank you!*
>
> Best,
> Authors

---

> > ### Comment · Reviewer_1ERZ · 2022-08-08
> > **Thanks for the response**
> >
> > Thanks to the authors for responding to my questions and comments. For missing details, it would be nice to explicitly mention that the details are in the supplementary.

---

> ### Author Response · Authors · 2022-08-02
> **Response to Reviewer 1ERZ (Part 1/2)**
>
> *We appreciate the positive and constructive comments from you! We have modified our paper according to your comments.*
>
> &nbsp;
>
> > **Q1: Firstly, it's not very clear what class of data the method operates on. Is it scenes or object classes? Only in the experiment section is this made clear (PartNet-Reasoning). This is an important detail that the paper ought to mention earlier.**
>
> &nbsp;&nbsp; &nbsp;Thank you for pointing it out! In our modified manuscript, we add one more line in the abstract (Line 18) to specify that our experiments are conducted on PartNet-Reasoning, and one more sentence in the introduction (Line 67-69) to introduce the PartNet-Reasoning dataset.
>
> > **Q2: The paper seems to introduce a depth-map conditioned neural field (line 123), but details and differences to conventional neural shape fields are missing.**
>
> * In our modified manuscript, we add more references to neural shape fields and discuss the differences in the related works (Line90-95).
> * We have in fact provided details about the neural descriptor field in Supplementary Material B.1. Specifically,
>
>     * For encoder, we use a PointNet-based encoder network with ResNet blocks. We use 5 resnet blocks, each with input dim 256 and output dim 128. At each step the output is concatenated with the input to produce the input of the next step. To enable communication between points at lower layers, we also add pooling and expansion layers between the ResNet-blocks. After the ResNet-blocks, the final output is pooled using max-pooling and then projected to a 128 embedding vector using a fully-connected layer.
>
>     * For decoder, we first concatenate the coordinates and the point cloud latent feature vector. The new input is passed through 5-layer resnet blocks. All the resnet blocks have a hidden size of 128. The features $\textbf{v}$ of the blocks are concatenated together and passed through a fully-connected layer to produce the 1-dimensional occupancy value. For color reconstruction, the features go through 3 linear layers with hidden size 128 and in the last layer, produce 3-dimensional r,g,b value. The batch size is 4. The learning rate is 1e-4. All embeddings have the dim 2049.
>
> > **Q3.1: what kind of parsing does the seq2seq model do?**
>
> * The seq2seq model performs semantic parsing, which transforms the natural language question into a program of neural operators which can be executed on the neural field.
> * We have in fact provided details about the seq2seq model in the Supplementary Material B.2. Specifically, Our semantic parser is an attention-based sequence to sequence (seq2seq) model with an encoder-decoder structure similar to that in [3]. The encoder is a bidirectional LSTM. The decoder is a similar LSTM that generates a vector from the previous token of the output sequence. Both the encoder and decoder have two hidden layers with a 256-dim hidden vector. We set the dimensions of both the encoder and decoder word vectors to be 300.
>
> > **Q3.2: Why were these three operators chosen?**
> * **From the perspective of benchmarks,** the *filter*, *query* and *count* operators are the prime operators proposed in previous visual reasoning datasets (*e.g.,* CLEVR[1], CLEVRER[2]), the compositions of these prime operators provide minimalist yet strong testbeds to evaluate the essential compositional and logical reasoning abilities of various models. Following these pioneering visual reasoning works, we also incorporate these prime operators into our dataset to evaluate the reasoning abilities of our proposed framework on object parts.
> * **From the perspective of methods,** we illustrate in the paper how these prime operators are implemented and incorporated into the descriptor field. State-of-the-art visual reasoning models[3,4] typically use supervised methods (*e.g.,* Mask-RCNN and attribute networks) to extract the object-centric representation. The *filter*, *query* and *count* operators are then executed on the object-centric representations. These object-centric representations usually require heavy supervision and can hardly generalize to novel objects. In our framework, we aim to seamlessly bridge the gap between perception (i.e. segmentation) and reasoning through neural field. In the paper, we demonstrate how the *filter*, *query* and *count* operators can be directly executed on the neural representations, and how the execution of these reasoning operators naturally brings out the object-centric representations.

---

### Official Review · Reviewer_owKC · 2022-07-12

**Rating:** 6
**Confidence:** 4
**Soundness:** 2 fair
**Presentation:** 2 fair
**Contribution:** 3 good

**Summary:**

This paper proposes a method for grounding concepts onto 3D representations inferred from images. The idea builds implicit 2d-to-3d methods like occupancy networks and neural descriptor fields, which encode an image to an embedding space, where, paired with a 3D query vector, the model returns a vector describing quantities of interest. This setup is paired with operators similar to "neuro-symbolic concept learner", which gives the model a way to answer questions about the input image. The model is first supervised for occupancy and color prediction, and then supervised to answer synthetic questions. The model is trained and evaluated on toy-ish data consisting of single objects on white backgrounds, where it outperforms some baselines.


**Questions:**

I asked a few questions in the Weaknesses part of my review.

**Limitations:**

I think the written limitations greatly understate the true limitations.

It says...
> while we show transfer results on real scenes

I don't think this was convincingly shown.

> an interesting direction of future work would be scale our framework to directly train on complex real-world scenes and ground more concepts from natural language on these real scenes is worth delving into in the future

This is so far beyond the paper's scope that it doesn't feel serious.




****
Post-rebuttal update: During the discussion phase the authors gracefully engaged with my complaints and toned down the contribution claims, so I am increasing my rating. I think people are generally interested in efforts to attach task-relevant features to neural fields, and this paper presents a reasonable step in that direction.

**Strengths And Weaknesses:**

I like the idea of using an intermediate 3D representation for the grounding task. I think this is a good idea. But I think that the difficult part here is in inferring the intermediate 3D representation for the grounding -- which is not this paper's focus. Instead, the paper is focused on toy scenes where 2D-3D lifting is relatively easy, and then combines neurosymbolic-style operators to perform (and supervise) some grounding tasks.

I am not very familiar with the baselines here, so I cannot judge the improvement over them.

A major factor in my negative impression for this paper is the overstatement of results. When a paper claims to have done something, when in fact it only achieved a version so simplified as to be trivial compared to the real issue, it feels frustrating to me as a reader.

The paper claims that segmentation emerges:
> "we observe the emergence of natural 3D segmentations of concepts, at the semantic and instance level, directly from the underlying supervision of downstream reasoning."

I don't understand why this claim is here at all. As explained later, semantic and instance segmentation are supervised through the querying task (line 192).


The paper claims that much of its training is self-supervised:
> "an intermediate representation should be discovered in a self-supervised manner"
> "descriptors are learned in a self-supervised manner, and implicitly capture the hierarchical nature of a scene"
> "After learning decoders Di through these self-supervised pre-training tasks..."

This is in reference to 3D shape and color reconstruction, from image input. This requires a 3D model or detailed multi-view reconstruction of the object created beforehand, before the method begins! (Not to mention it requires object-centric data carefully collected by a human.)


The paper claims that it generalizes to unseen categories and real scenes:
> "It can also generalize well to unseen categories and real scans."
> "our 3D-CG trained on seen categories can be generalized directly to unseen categories and real scans"
> "estimate 3D-CG’s ability to generalize to real scenes"

In fact, the experiment here completely abstracts away the model's ability to infer a 3D representation from a 2D image, and instead queries the model's segmentations at ground-truth voxels. As for the categories, the results on the synthetic car and bed in Fig4b are quite unusual -- ignoring the fact that the blue wheel is in the wrong place, how did the model know to estimate the occluded side of the car and put 4 wheels, when only 2 are visible? Was 3D ground truth provided here as well?

---

> ### Author Response · Authors · 2022-08-02
> **Response to Reviewer owKC (Part 2/2)**
>
> **3.Real-world Extension**
>
>  &nbsp; &nbsp; We show that our framework can be directly applied to more diverse and complex real-world scenes on our demo website (https://sites.google.com/view/3d-cg/home). Same as our paper, we can learn the features through neural descriptor fields (NDF) and ground the concepts with question-answering pairs. The concept embeddings can be initialized using CLIP language embeddings, and the NDF features can be volumetrically rendered in each multiview image to align with the CLIP embeddings. The resultant 3D descriptor field and concept embeddings can be directly integrated with our proposed operators.
>
> &nbsp;
>
> [1] Self-Supervised Scene Representation Learning. Vincent Sitzman. 2020
>
> [2] Neural Descriptor Fields: SE(3)-Equivariant Object Representations for Manipulation. Anthony Simeonov, Yilun Du, Andrea Tagliasacchi,
> Joshua B. Tenenbaum, Alberto Rodriguez, Pulkit Agrawal, Vincent Sitzmann. ICRA2022.
>
> [3] Neural-Symbolic VQA: Disentangling Reasoning from Vision and Language Understanding. Kexin Yi*, Jiajun Wu*, Chuang Gan, Antonio Torralba, Pushmeet Kohli, Joshua B. Tenenbaum. NeurIPS2018.
>
> [4] The Neuro-Symbolic Concept Learner: Interpreting Scenes, Words, and Sentences From Natural Supervision.  Jiayuan Mao, Chuang Gan, Pushmeet Kohli, Joshua B. Tenenbaum, Jiajun Wu. ICLR2019.
>
> &nbsp;
>
> *We wish that our response has addressed your concerns, and turns your assessment to the positive side. If you have any more questions, please feel free to let us know during the rebuttal window. Thank you very much! We appreciate your suggestions and comments! Thank you!*
>
> Best,
> Authors

---

> > ### Comment · Reviewer_owKC · 2022-08-05
> > **Discussion**
> >
> > Thank you for the detailed rebuttal. There is a lot to process here. For me, the thing of paramount importance is: whether the paper does what it claims. One way to address this is by doing more (which you've done for some parts), and another way is to adjust the claims slightly (which you've also done, for other parts). I still have a few gripes, which I hope you will discuss with me.
> >
> > > We agree that the term self-supervised may not accurately describe our method, as it requires both color and occupancy data, though such learning of neural fields has been widely referred to as self-supervised in prior literatures [1,2]. [...] To address this concern, we have revised the term self-supervised to weakly-supervised in our revised manuscript.
> >
> > Great. Weakly-supervised fits your setting much better, not only because of the direct 3D supervision, but also because the descriptors receive supervision from the VQA.
> >
> > > [3,4] require supervision to obtain object-centric representations. Instead, 3D-CG can ground the concepts without any pre-segmentations.
> >
> > This does not seem accurate. 3D-CG's input data is clearly pre-segmented and object-centric. Right? It's ShapeNet objects on white backgrounds.
> >
> > > we refer to reasoning as the process of looking at RGBD images and reasoning about paired questions and answer, as stated in the first sentence in the abstract
> >
> > I think you're saying that the first sentence defines what you mean by "reasoning", but the first sentence right now says "we address the challenging problem of 3D concept grounding (i.e. segmenting and learning visual concepts) by looking at RGBD images and reasoning about paired questions and answers". This doesn't give a definition for reasoning. If your definition for reasoning is VQA, I think you should call it VQA every time. In image-based VQA, people are very up-front about the distance between VQA and "reasoning" [5].
> >
> > > we control the reconstruction part and show the segmentations on ground-truth voxels for fair comparison
> >
> > This is fair, and thank you for the new tables. I think the more alarming part (relating again to a mismatch between claims and results) is Figure4b, apparently showing "generalization to new categories". I think you used ground-truth 3D segmentation to produce the visualizations in that figure, and I think doing so without clearly acknowledging it in the figure or its caption, is bad.
> >
> > > extension [with CLIP embeddings and so on]
> >
> > I don't know what to make of this right now. It's kind of cool, but I don't know exactly how you did it, and it doesn't seem like it's part of the paper.
> >
> > Another issue, related to VQA and weak supervision, and not yet addressed (I think), is this idea of segmentation "emerging". I think this claim is warranted in the sense that you can show segmentation outputs, without using segmentation labels (thanks to the VQA). But the paper sometimes suggests more than this. Here are some examples:
> >
> > > "Semantic and instance segmentations naturally emerge with 3D-CG." "the segmentations emerge naturally from scratch without any restrictions." "not only can semantic and instance segmentations emerge naturally, but visual reasoning can also be well performed."
> >
> > These sentences ask a reader to believe that segmentation is emerging in a way that is detached from the VQA supervision, which is not true. For an alternative way to talk about such results, citation 47 ("GroupViT: Semantic segmentation emerges from text supervision", CVPR 2022) might be a good reference.
> >
> > A subtler (but much more frequent) issue is: the use of the word "naturally". It doesn't seem like this word is adding anything concrete. Do you mean something specific when you use it? Right now, I think you could delete almost all instances of "naturally", and the paper would get stronger.
> >
> >
> >
> > [Continued refs:]
> >
> > [5] "How Transferable are Reasoning Patterns in VQA?" CVPR 2021

---

> > > ### Author Response · Authors · 2022-08-06
> > > **Response to Reviewer owKC (Part 2/2)**
> > >
> > >
> > > **[*Visual Quesiton-Answering (VQA)* and *Reasoning*]**
> > >
> > > Thanks for bringing up this question.
> > > * We argue that *VQA* and *reasoning* are not mutually-exclusive terms. Reasoning is an **ability**, while VQA is a **task** and a **testbed** which can evaluate this ability. As stated in [5], *"Visual Question Answering (VQA) in particular has become a testbed for the evaluation of the reasoning"*.
> > > * For the definition of *reasoning* in our paper, we refer to [6,8] and define it as *the utilization of **compositional** operators to perform complex visual question answering tasks that go beyond pattern recognition and bias exploitation, such as counting, comparing, and logical reasoning*.
> > > * We have added the definition accordingly in Line 1 of the modified paper.
> > >
> > >
> > > **[*emerging* and *natural*]**
> > > * We have added the phrase *from question-answering supervision* after *emerge* for Line 312, 332 and the caption of Figure 2.
> > > * We have deleted all *natural* and *naturally* in our revised version. Thank you for the advice!
> > >
> > >
> > > &nbsp;&nbsp;
> > >
> > > [3] Neural-Symbolic VQA: Disentangling Reasoning from Vision and Language Understanding. Kexin Yi et al. NeurIPS2018.
> > >
> > > [4] The Neuro-Symbolic Concept Learner: Interpreting Scenes, Words, and Sentences From Natural Supervision. Jiayuan Mao et al. ICLR2019.
> > >
> > > [5] How Transferable are Reasoning Patterns in VQA? Corentin Kervadec et al. CVPR 2021.
> > >
> > > [6] CLEVR: A Diagnostic Dataset for Compositional Language and Elementary Visual Reasoning. Justin Johnson et al. CVPR 2017.
> > >
> > > [7] NeRF: Representing Scenes as Neural Radiance Fields for View Synthesis. B. Mildenhall et al. ECCV 2019.
> > >
> > > [8] From machine learning to machine reasoning. Leon Bottou. Machine Learning 2014.
> > >
> > > &nbsp;&nbsp;
> > >
> > > *Thanks again for your insightful suggestions. We would really appreciate it if you could **raise your rating**. Please do not hesitate to contact us if there are other clarifications or experiments we can offer.*
> > >
> > > *Thank you for your time!*
> > >
> > > Best,
> > > Authors

---

> > > > ### Comment · Reviewer_owKC · 2022-08-07
> > > > **Thanks**
> > > >
> > > > I think the paper is a lot better now. I've updated my rating. Thanks for the discussion.

---

> > > ### Author Response · Authors · 2022-08-06
> > > **Response to Reviewer owKC (Part 1/2)**
> > >
> > > *We are truly thankful for your constructive comments, which significantly  strengthen our work. We provided a detailed response below, and also revised the paper accordingly.*
> > >
> > > &nbsp;
> > >
> > > **[Pre-segmentations]**
> > >
> > > * Sorry for the confusion. We actually mean that when applying neuro-symbolic visual reasoning frameworks [3,4] to concept grounding, a pre-segmentation model like Mask-RCNN is utilized to propose **part** segmentations or part-centric representations for question answering. Our model aims to resolve that by learning from question-answer pairs on neural field, without using pre-annotated part segmentations.
> > >
> > > * Additionally, we would like to point out that previous neuro-symbolic visual reasoning frameworks [3,4] work on simple scenes composed of elementary objects (*e.g.,* CLEVR [6] scenes are composed of objects like cubes and spheres, with white background), while 3D-CG works on our proposed PartNet-Reasoning dataset. For PartNet-Reasoning, each scene only has one object, but the complex part-whole structures and part-based concepts of different object categories make the segmentations harder and the question-answering pairs richer. We discussed this in Line 226 of the paper, "Instead of object-based visual reasoning [6,3,4] where objects are spatially disentangled, which makes segmentations quite trivial, we seek to explore part-based visual reasoning where segmentations and reasoning are both harder."
> > >
> > >
> > > **[Ground-truth Voxels]**
> > >
> > > Thank you for pointing it out! We have added it to the caption of Figure 4.
> > >
> > > **[Extension to Real-World Scenes]**
> > > * The real-world experiments are in response to the comment:
> > > > *"an interesting direction of future work would be scaling our framework to directly train on complex real-world scenes and ground more concepts from natural language on these real scenes"*
> > >     *This is so far beyond the paper's scope that it doesn't feel serious.*
> > >
> > >     The real-world experiments show that we indeed feel serious about this perspective and have been working on it, as we claimed in our *future work* part. We aim to show that the experiments on the synthetic dataset could lay the foundation for future attempts on real-world scenes like this. Below we provide the implementation details on the new real-world results.
> > >
> > > * To obtain our real-world results, we similarly construct a Neural Descriptor Field (NDF) which maps each xyz coordinate in $\mathbb{R}^3$ to a corresponding high dimensional descriptor in $\mathbb{R}^{d}$. However, instead of obtaining the descriptor field using a trained occupancy network (since we do not have ground truth occupancy), we explicitly train this descriptor field using volumetric rendering. In particular, Similar to Neural Radiance Field (NeRF) [7], we render a 512-dim feature vector in addition to the rgb values along the rays in each pixel in a set of multi-view images, where we seek to match the RGB value of each pixel, as well as features of a CLIP model at each associated pixel. In this way, we infer a dense neural 3D descriptor field from the alpha compositing operation.
> > > * Once we have this set of 3D neural descriptors, we may likewise apply each of our proposed reasoning operators (*e.g.,* filter, query, and counting) to the descriptor field. While the precise method to obtain the Neural Descriptor Field is different from the one used in the paper, our results indicate that our proposed *concept grounding on neural field* framework may be more widely applied to realistic scenes.
> > >
> > > * We also added one sentence in Line 339, *"A promising direction of future work would be to explore the combination of  large pre-trained visual-language models and volumetric rendering on multi-view images."*.

---

> ### Author Response · Authors · 2022-08-02
> **Response to Reviewer owKC (Part 1/2)**
>
> *We thank the reviewer for the constructive comments. We have modified our paper according to these comments.*
>
> &nbsp;
>
> **1.Clarifications**
>
> We would like to clarify the following statements about the content of the paper.
>
> * **[Self-Supervised Learning of the Neural Field]**
>     * We agree that the term *self-supervised* may not accurately describe our method, as it requires both color and occupancy data, though such learning of neural fields has been widely referred to as self-supervised in prior literatures [1,2].
>     * To address this concern, we have revised the term *self-supervised* to *weakly-supervised* in our revised manuscript.
>     * We would like to further emphasize that the claim on supervision is based on comparisons with other neural-symbolic reasoning methods (*e.g.,* NS-VQA[3] and NSCL[4]). These methods require supervision to obtain object-centric representations. Instead, 3D-CG can ground the concepts without any pre-segmentations.
>
> - **[Segmentation through Downstream Reasoning]**
>
>     * In our paper, we refer to *reasoning* as the process of *looking at RGBD images and reasoning about paired questions and answer*, as stated in the first sentence in the abstract. Since our approach utilizes visual question operators such as *query* to obtain the object and instance segmentation, we thus state that we obtain segmentations directly from downstream reasoning.
>     * To make this distinction more clear, in our revised manuscript, we have changed the word *reasoning* to *visual question answering* in Line 72.
>
>
> - **[Inferring Intermediate 3D Representations]**
>
>     * We agree that inferring 3D representations for grounding from 2D images is indeed a challenging problem.
>     * As stated in the introduction of our paper, we propose and argue that *neural fields serve as natural intermediate 3D representations of a 2D image for concept grounding*. In contrast to traditional 3D representations such as point clouds and meshes, neural fields continuously encode correspondences in 3D, enabling us to segment and learn visual concepts flexibly.
>
>
> - **[Inferring the occluded parts]**
>
>     In the reconstruction task, the model is trained to predict the voxel occupancy values, and learns to reconstruct the full 3D shape given only partial and single-view images based on symmetry priors. The model trained on *Cart* learns to estimate the occluded side and reconstruct all 4 wheels. The same ability can be transferred to the prediction of the *Car*.
>
> &nbsp;
>
> **2.Additional segmentation results based on predicted voxels**
>
> &nbsp;&nbsp;&nbsp;3D-CG first reconstructs the 3D shape from images and then performs reasoning in the 3D space. However, for fair comparison with baselines for the segmentation task, we control the reconstruction part and use the ground-truth voxels **for the segmentation evaluations only**. Here, we provide additional experimental results when the segmentations are performed on the reconstructed voxels.
> * **[additional experimental results]**
>
>     We add one experiment where we show the *segmentation results on predicted voxels from reconstruction*. Specifically, the neural descriptor field predicts the occupancy values of the sampled 3D coordinates, and reconstructs the 3D shape by keeping all the 3D coordinates with occupancy values that are greater than 0.5. The results are added to the supplementary material section D as well as shown below.
>
> Table A. Semantic segmentation IOU based on predicted voxels
> |          | Chair | Table | Bag | Cart | Mean |
> | -------- | ---- | ---- | ---- | ---- | ---- |
> | CVX | 38.1 | 49.5 | 40.3 | 26.0 | 38.5 |
> | CVX+L | 34.0  | 39.0 | 38.3 | 29.2 | 35.1 |
> | BAE | 36.3 | 50.1 | 50.0 | 38.5 | 43.8 |
> | BAE+L | 41.1 | 51.6 | 51.7 | 34.5 | 44.8 |
> | 3D-CG | 69.3 | 65.7 | 56.9 | 48.2 | 60.0 |
>
> Table B. Instance segmentation IOU based on predicted voxels
> |          | Chair | Table | Bag | Cart | Mean |
> | -------- | ---- | ---- | ---- | ---- | ---- |
> | CVX | 29.4 | 26.4  | 18.2 | 13.4 | 21.9 |
> | CVX+L | 22.9 | 28.2 | 22.6 | 13.8 | 21.9 |
> | BAE | 38.7 | 22.7 | 28.1 | 24.8 | 28.6 |
> | BAE+L | 37.3 | 34.5 | 25.1 | 26.7 | 30.9 |
> | 3D-CG | 57.1 | 58.7 | 42.0 | 35.4 | 48.3 |
>
> * **[controlled study for fair comparison]**
>
>     We still believe that the experiment in the main paper is a meaningful controlled study for fair comparison. It's useful to provide an oracle reconstruction for the evaluation of segmentations performances alone. Since our method performs much better than the baselines on 2D->3D reconstruction, we control the reconstruction part and show the segmentations on ground-truth voxels **for fair comparison**.

---

### Author Response · Authors · 2022-08-02
**General Response to All Reviewers**

We thank all the reviewers for the insightful comments and constructive suggestions to strengthen our work. In addition to the response to specific reviewers, here we would like to highlight our contributions and the new experiments that we add in the rebuttal.

**1.Our Contributions**

We are glad to find out that the reviewers generally acknowledge our contributions:
* We propose to use neural field as the intermediate 3D representation for the grounding task [owKC, 1ERZ, gaS1, Xamz] without any explicit supervision [1ERZ, gaS1]. This makes the reasoning process continuous and offers previously unseen differentiability properties [1ERZ].
* This paper proposes some novel operators, particularly for counting which has never been proposed elsewhere [1ERZ].
* Experimental results show superior performances over baselines [gaS1, Xamz, 1ERZ].

**2.New Experiments**

In this rebuttal, we have added more supporting experiments to address reviewers’ concerns.

* **[Segmentation results based on predicted reconstruction]** We add segmentation performances based on predicted reconstruction instead of ground-truth voxels in Section D of Supplementary Material to address the concern of reviewer [owKC].
* **[Quantitative results of generalization]** We add quantitative results of generalization to novel categories, as required by reviewer [gaS1].

* **[Extension to more diverse real-world scenes]** To address the concerns of the reviewers [owKC, gaS1, Xamz], we show that our framework can be directly applied to more diverse and complex real-world scenes. We may learn neural descriptor fields directly across multiview real images by training the descriptor field to volumetrically render CLIP features in each multiview image. The resultant 3D descriptor field can be directly integrated with our proposed operators as illustrated on our demo website (https://sites.google.com/view/3d-cg/home).


We hope our responses below convincingly address all reviewers’ concerns. We thank all reviewers’ time and efforts again!

---

### Author Response · Authors · 2022-08-09
**Summary of our rebuttal and discussion**

*We genuinely thank all reviewers and ACs for their efforts and time in reviewing our paper, as well as their constructive suggestions that contribute to the improvement of our paper! We sincerely appreciate the positive 6-8-6-6 evaluation from reviewers owKC, 1ERZ, Xamz, gaS1.*

&nbsp;

Here is a summary of our responses:

**Contributions**

We would like to emphasize the contributions of this paper:
* we propose 3D-CG, which utilizes the differentiable nature of neural descriptor fields (NDF) to ground concepts and perform segmentations
* We define a set of neural operators, including a neural counting operator on top of the NDF
* Semantic and instance segmentations can emerge from question answering supervision
* Our 3D-CG outperforms baseline models in both segmentation and reasoning tasks

**Additional Experiments**

* **[segmentation]** To address the concern of reviewer [owKC], we add additional experimental results of segmentation on predicted voxels.
* **[generalization]** To address the concern of reviewer [Xamz], we add quantitative results of generalization to novel categories.
* **[real-world experiments]** To address the concerns of the reviewers [owKC, gaS1, Xamz] and show that 3D-CG can be combined with large pre-tained vision-language model and NERF for real-world scenes. The premilinary results are shown here https://sites.google.com/view/3d-cg/home.

**Writing**

We thank reviewer owKC and reviewer 1ERZ for the helpful writing suggestions.  We have made the following efforts to avoid overclaiming and make our paper more readable:
* We replace words that may cause overstatement or misunderstanding (*e.g.*, self-supervised, emerge naturally *etc*). [owKC]
* We add definitions for *reasoning* and *question-answering* at the beginning of our paper. [owKC]. We specify that we are working on our proposed PartNet-Reasoning dataset at the beginning of our paper. [1ERZ]
* We add significantly more related works. [1ERZ]
* We add details of 3D-CG in our paper as well as in the supplementary material. [1ERZ]

&nbsp;

*We owe many thanks to the reviewers for their insightful suggestions which help improve our paper a lot. The additional experiments and modifications on language have been delivered in our paper and will be reflected in the final version as well.*

Best,
Authors

---

### Meta-Review · Area_Chair_2XMG · 2022-08-26

**Recommendation:** Accept
**Confidence:** Certain

**Metareview:**

The paper received all positive reviews (3x weak accept ratings, 1x strong accept rating). The meta-reviewer agrees with the reviewers' assessment of the paper.

**Award:**

No

---

### Decision · Program_Chairs · 2022-09-14

Accept